# Investigation of the Effects of Pulse-Atomic Force Nanolithography Parameters on 2.5D Nanostructures’ Morphology

**DOI:** 10.3390/nano12244421

**Published:** 2022-12-11

**Authors:** Paolo Pellegrino, Isabella Farella, Mariafrancesca Cascione, Valeria De Matteis, Alessandro Paolo Bramanti, Antonio Della Torre, Fabio Quaranta, Rosaria Rinaldi

**Affiliations:** 1Department of Mathematics and Physics “Ennio De Giorgi”, University of Salento, Via Monteroni, 73100 Lecce, Italy; 2Institute for Microelectronics and Microsystems (IMM), CNR, Via Monteroni, 73100 Lecce, Italy; 3STMicroelectronics S.r.l., System Research and Applications (SRA) Silicon Biotech, Lecce Labs, Via Monteroni, 73100 Lecce, Italy

**Keywords:** AFM-based nanofabrication, atomic force-nanolithography, pulse-atomic force nanolithography, atomic force microscopy

## Abstract

In recent years, Atomic Force Microscope (AFM)-based nanolithography techniques have emerged as a very powerful approach for the machining of countless types of nanostructures. However, the conventional AFM-based nanolithography methods suffer from low efficiency, low rate of patterning, and high complexity of execution. In this frame, we first developed an easy and effective nanopatterning technique, termed Pulse-Atomic Force Lithography (P-AFL), with which we were able to pattern 2.5D nanogrooves on a thin polymer layer. Indeed, for the first time, we patterned nanogrooves with either constant or varying depth profiles, with sub-nanometre resolution, high accuracy, and reproducibility. In this paper, we present the results on the investigation of the effects of P-AFL parameters on 2.5D nanostructures’ morphology. We considered three main P-AFL parameters, i.e., the pulse’s amplitude (setpoint), the pulses’ width, and the distance between the following indentations (step), and we patterned arrays of grooves after a precise and well-established variation of the aforementioned parameters. Optimizing the nanolithography process, in terms of patterning time and nanostructures quality, we realized unconventional shape nanostructures with high accuracy and fidelity. Finally, a scanning electron microscope was used to confirm that P-AFL does not induce any damage on AFM tips used to pattern the nanostructures.

## 1. Introduction

In the last decades, growing demand for nanostructures has given the academic and scientific community the exciting challenge of the development of alternative, easy-to-use, and effective nanofabrication technologies, bringing us into the nano age. Despite the recent, numerous research efforts in this area, the challenge to fabricate nanostructures with increasingly smaller sizes and in a controllable way remains open.

Recently, owing to the high resolution, versatility, and reproducibility, Atomic Force Microscopy (AFM)-based nanofabrication techniques emerged as one of the most prominent nanofabrication approaches. Since its discovery in 1986 by Binnig et al. [1], AFM has been widely adopted: initially for sample surface investigation down to the atomic resolution [2,3], and lately for nanolithography [4,5] and nanofunctionalization [6] purposes. These nanofabrication approaches, generally known as Scanning Probe Lithography (SPL) [4,7], are based on the use of AFM probes to directly fabricate nanostructures on the sample surface through various mechanisms, opening up a wide range of possible applications [8]. Compared to more conventional top-down fabrication techniques, such as those based on Electron Beam Lithography (EBL) [9,10], Focused Ion Beam (FIB) [11,12,13,14,15], or Ultra-Violet lithography (UV Lithography) [16], TBN techniques are cheaper, more flexible, environment-friendly and maskless, and target more materials [4]. Moreover, the same cantilever used for fabrication could be employed to image the structures immediately afterward [17].

To date, the main SPL techniques developed include Local Anodic Oxidation (LAO) [18,19,20], dip-pen [6,21,22], thermal–mechanical writing [23,24], electric nanolithography [25], and mechanical nanolithography (m-SPL) [26,27,28]. In particular, in m-SPL, the AFM tip patterns the surface by means of three different operation modes: nanoindentation (NI), Static Plowing Lithography (SPL) or Dynamic Plowing (DPL), which descend from indentation [29,30,31], contact [32,33], and tapping scanning modes [34,35,36,37], respectively. The m-SPL techniques have been employed to pattern 1D and 2D nanostructures [4,5], including nanogrooves [35,38], nanochannels [37,39], arrays of nanodots [34,40], and 3D shapes [41,42]. To improve the quality of the 2D and 3D nanostructures, additional support energy sources, such as ultrasound [43,44], high tip temperature [23,45], or rotation [46], are required. Resorting to those support sources inevitably makes the fabrication process slower and more complex. Moreover, more sophisticated types of equipment are required and only specific materials can be targeted [47]. In order to overcome the drawbacks of the conventional m-SPL techniques, we have already proposed a new AFM-based nanolithography technique, termed Pulse-Atomic Force Lithography (P-AFL) [48], as an evolution of AFM nanoindentation.

In our previous paper [48], we demonstrated the effectiveness of P-AFL in the fabrication of nanochannel with a continuously varying depth profile on a thin polymer layer. The obtained results suggest the tremendous potential of P-AFL in the building of 2.5D structures. P-AFL offers numerous advantages with respect to the conventional m-SPL techniques, including the possibility to easily tune the depth of the nanostructures to pattern the material surface with high accuracy and nanometric spatial resolution. In addition, our technique is less expensive with respect to other m-SPLs, since the manufacturing process does not induce wear on the AFM tip, and it does not require sophisticate nanolithography instrumentation. As for the other m-SPL techniques, the formation of pileups at the borders of nanostructures is observed, due to the mechanical movement of polymer from the groove during the P-AFL procedure. To overcome this drawback, an effective and easy method was proposed to eliminate polymer pileups, paving the way to the integration of these nanostructures in nanodevices [49].

The P-AFL technique is based on the indentation mode of operation, and by appropriately setting both the amplitude and width of the voltage pulse trains applied to the piezo scanner, as well as the distance between following nanoindentations, it is possible to realize nanochannels with high resolution in the xy-plane with a continuously variable depth profile (z-axes) in a single pass.

Therefore, the quality of the patterned nanostructures and the reproducibility of the proposed nanolithography process depend on values set for the main pulse-AFL parameters: the pulse amplitude (setpoint), the pulse width, and distance between following indentations (step). With this aim, in the present work we investigate the effect on depth, width, and overall shape of the nanochannels by the systematic variation of aforementioned P-AFL parameters.

After this step, we used the optimized set parameter value to obtain nanostructures with unconventional shape, such as a set of concentric circles, nanolabyrinth, and serpentine-like geometry having constant depth. In addition, triangular, circular, and serpentine-shape grooves with linearly increasing depth profile were fabricated. Finally, the tip wear was investigated using Scanning Electron Microscopy (SEM) to observe the AFM probes before and after numerous lithography tests.

## 2. Materials and Methods

### 2.1. Substrate Fabrication

A 4-inch standard (100) Silicon wafer with an 0.5 µm thermally grown silicon dioxide layer was used as the substrate for the P-AFL test. The silicon wafer was firstly cleaved in 1.5 × 1.5 cm^2^-wide pieces; successively, they were cleaned by sequential sonication bath in acetone and 2-propanol for 15 min each at a temperature of 180 °C on a hotplate. Both acetone (99.5%) and 2-propanol (99.5%) were purchased from Sigma-Aldrich. Successively, samples were rinsed under flowing deionized water and finally dried out under a nitrogen gas stream for about 30 s. To promote resist adhesion, the silicon substrates were further dehydrated on a hotplate for 15 min at 200 °C before resist spin coating. Successively, a thin layer of Polymethyl Methacrylate (PMMA) (950,000 Da, in solvent anisole, purchased by MicroChem), was spun at 4000 rpm for 30 s on Silicon substrates using a semiautomatic spinner DELTA 80T (SUSS MicroTec Corp, Garching, Germany), and baked on a hotplate at 180 °C for 90 s for solvent dry-out. PMMA layer thickness, measured with an Alpha-Step P6 profilometer (KLA-Tencor Corporation, Milpitas, CA, USA), was about 190 nm.

### 2.2. Instrumentation for Nanopatterning and Characterization

The nanogroove fabrication and subsequent characterization was carried out at ambient conditions (room temperature of about 22 °C and relative humidity less than 50%) with a commercial AFM NTEGRA provided by NT-MDT Co. (NT-MDT Spectrum Instruments, Moskow, Russia). The AFM NTEGRA is equipped with a double feedback system: one strictly controls the movements of the piezo scanner in the xy plane while the other controls the z direction. Moreover, the instrument is provided with built-in scanner displacement sensors that can trace and control the displacement of the scanner and, therefore, compensate imperfections in its movement such as non-linearity, bow, creep, and hysteresis, as schematically reproduced in Figure 1a). These feedback systems and sensors allow for working with sub-nanometric precision, which is particularly valuable when carrying out nanomanipulation and nanolithography. The AFM is also equipped with an oscilloscope and a nanolithography modulus, which enable force spectroscopy measurement and nanolithography testing, respectively.

Doped, diamond-coated conductive probes were used in contact mode to perform the nanolithography tests. DCP20 probes are V-shaped cantilevers, with conical tips 10–15 µm high at the apex with a cone angle less than 22°. The typical curvature radius is approximately 100 nm. The nominal spring constant (*k*) is 80 N/m, and the resonance frequency is 420 kHz. Prior to P-AFL experiments, the calibration of the DPC20 cantilevers was carried out via the thermal noise method [50] and the relative force constant was determined to be 81.2 ± 1.6 N/m.

NSG01 AFM tips were used in semi-contact mode to characterize at high-resolution the pristine and patterned PMMA. NSG01 probes present a sharp pyramidal tip with a curvature radius between 6 and 10 nm, mounted at the apex of a silicon rectangular cantilever. The NSG01 tips have a typical resonance frequency of about 150 kHz and a nominal spring constant of 5.1 N/m. All the used probe tips were provided by NT-MDT Spectrum Instruments, Moskow, Russia.

Each topographical image was digitally treated with a second-order plane fit and with a second-order flattening to suppress both the tridimensionality and bow effects. Then, the topography images of the nanochannels, patterned with different combinations of P-AFL parameters, were also analysed to quantify the nanogrooves’ geometrical parameters. In detail, the depth, width, and roughness parameters were calculated as mean value ±SD by practicing 20 cross-sections per acquisition.

### 2.3. Pulse-AFM Lithography

The technique used for our nanolithography experiments, Pulse-AFL, was first presented in our previous papers [48,49]. Therefore, it is worth briefly remarking on its fundamental aspects to understand the experiments reported in the present work and point out the effective capabilities of the technique.

P-AFL is a contact mode AFM-based nanolithography technique in which the sample surface is patterned by a series of closely spaced tip indentations corresponding to each voltage pulse applied to a piezo-scanner (Figure 1b). The characteristics of the voltage pulses and the distance between indentations are the key parameters of the P-AFL and their optimization drives the results of the nanolithography process. Different from the DPL method, in which the AFM probe is not in contact with the sample surface and sculpts it vibrating, in P-AFL the tip is in contact with the surface and is forced to indent it. In this mode, a wider range of forces (from few µN to tens of µN) can be applied, allowing to pattern deeper nanogrooves (tens of nm). In addition, P-AFL does not present the DPL’s problems such as polymer residues attached on the tip or polymer debris falling into the patterned channels.

In more detail, in P-AFL triangular voltage pulses of a certain duration and constant amplitude are fed to the scanner, which moves in z-direction, forcing the AFM tip to indent the sample surface. The amplitude (setpoint) and the width of the pulse are correlated to the force applied to indent the nanostructures and to the duration of nanoindentation, respectively (Figure 1b). In addition, it is possible to vary the distance between two following indentations (step), from few nanometres up to a few hundred micrometres, which affects the continuity of the pattern (Figure 1b,c). Consequently, the choice of the proper pulse parameters values as well as indentation step enable the optimization of the desired structures. Moreover, in P-AFL the tip indents the surface moving along a direction orthogonal to the cantilever’s main axis (Figure 1d).

Finally, P-AFL technique comes in two variants, termed Constant Pulse- and Gradient Pulse-AFL (CP- and GP-AFL, respectively), which mainly differ in the amplitude of the pulses applied to the scanner: in CP-AFL the pulses’ amplitude is constant while in GP-AFL it can vary within a fixed setpoint range. Then, by means of CP- and GP-AFL, is it possible to pattern nanostructures with constant and gradient depth profile, respectively.

### 2.4. Force Spectroscopy

P-AFL is critically based on the control of the force applied to the tip during indentation process. Thus, before nanolithography experiments, the forces acting on the tip during the nanolithography procedure were estimated by force-spectroscopy analysis.

When the probe is in contact with the surface, any variation in the signal applied to the z-contact of the scanner (the *Z* signal) causes the cantilever to rise; then, a proportional modification of deflection (DFL) signal occurs. Using the DFL(*Z*) function and assuming that the cantilever stiffness is known, it is possible to calculate the forces acting on the probe when a certain setpoint value is assigned. In particular, by Hooke’s law:F = k·ΔZ (1)
where k is the cantilever stiffness (N/m) and ΔZ is the scanner height (nm) [51]. The real cantilever stiffness was estimated via the thermal noise method [50,52,53].

Therefore, from the force–distance curves the cantilever deflection is measured as a function of the piezo scanner extension. In order to permit to the AFM tip to penetrate through the substrate, the force applied must exceed a certain threshold value, which is at least of a few micronewtons in AFM nanolithography and is strictly correlated to substrate composition and structure [54]. The force-spectroscopy analysis was performed on fourteen setpoint values, ranging from 0.5 nA to 7 nA in steps of 0.5 nA. Those setpoints were successively used in all the tests carried out.

### 2.5. Software

The topographic AFM images, the measurement of the force acting on the tip during lithography by force-spectroscopy, and all the nanolithography tests were performed by means of NOVA_PX software, whereas their analysis was performed using Image Analysis-P9 (IA-P9) software. IA-P9 and NOVA_PX were both obtained from NT-MDT Spectrum Instruments, Moskow, Russia. The experimental data were analysed and plotted by using Origin Pro v8 (Origin-Lab Corporation, Northampton, MA, USA). The SEM images were analysed by means of ImageJ software [55].

### 2.6. Statistical Analysis

The results report in this work are shown as mean values and associated standard deviation (±SD). The difference among data was analysed through ANOVA multiple comparisons. The differences were statistically significant when *p* < 0.01.

## 3. Results

The P-AFL experiments reported in this paper were performed on a thin and smooth PMMA layer (Figure 2a,b), whose root-mean-square surface roughness (R_q_), calculated from the AFM topographic acquisitions over fifteen areas (5 × 5) µm^2^ wide, corresponds to (0.45 ± 0.09) nm. In addition, the thickness of PMMA was measured using a profilometer, with results equal to ~70 nm.

Preliminarily, the normal force exerted by the tip on the surface in correspondence of each setpoint was estimated in contact mode by averaging over 25 force–distance curves, acquired with DCP20 tip on a silicon substrate. The averages and standard deviation of the forces calculated are reported in the Table 1.

Three main parameters of the P-AFL process, such as the force acting on the AFM probe (setpoint), the distance between following indentations (step), and the pulse amplitude (briefly, pulse), were identified as key parameters for the Pulse-Atomic Force Lithography technique. These parameters were then systematically modified to obtain nanostructures with desired depth (d), regular shape and width (w), and in the shortest time possible. To this aim, several nanolithography tests were carried out on a thin PMMA film, by changing time by time, one of the three parameters and keeping the other two fixed. The patterns designed as test structure were nanochannels; their depths, widths and inner flatness were carefully measured and analysed for each nanolithography test to understand how voltage pulse characteristics may affect the final structures. In the following subsections, the results obtained in the different testes are presented and discussed.

### 3.1. Impact of the Indentation Step on the Nanogrooves Shape

As described above, the P-AFL technique is based on nanoindentation; therefore, to obtain nanogrooves with continuous profile, the distance between successive indentations, defined as step, is a critical P-AFL parameter. As long as the step was kept as low as possible (10 nm), we were able to realize nanochannels with continuous and homogeneous profiles, and low roughness [48]. Nevertheless, with steps so short, the tip must perform a high number of indentations for patterning a groove with a certain length, which entails a significant increase in the nanolithography time. To reduce the lithography duration, the change in the nanogrooves’ morphology as a function of indentation distance has been investigated by patterning a set of fourteen parallel lines, each line corresponding to a certain step. The template was generated by means of Nova_Px software.

All the nanogrooves in the template were patterned with a fixed setpoint and pulse width corresponding to 5 nA (9.44 ± 0.25 µN) and 10 ms, respectively, while the step was increased from 10 nm to 100 nm, in a step of 10 nm, and from 100 nm to 200 nm in a step of 25 nm. The nanolithography process was steadily monitored by an oscilloscope, and the AFM probe deflection signals during the patterning procedure were reported in Appendix A. The nanolithography results were successively characterized with AFM using sharp tips at high resolution. The AFM topography images of the channels sculpted with increasing steps are presented in Figure 3a–c. At first glance, for step up to 50 nm, the lines appear continuous and contoured by asymmetrical pileups [48,49]; whereas for higher step values they start to be dotted, and, in correspondence to the highest steps (from 125 nm to 200 nm), single indentations can be well-recognized. In this case, every single hole (nanoindentation) is surrounded by a PMMA ring bulge. Further insight into the changes in line morphology can be obtained by inspecting the nanogroove cross-sections (Figure 3d–i) (Appendix A). Three main behaviours of the morphology can be recognized as the step increases: (i) for steps up to 40 nm, the groove profiles are very smooth and continuous; (ii) from 50 nm to 70 nm step, a zig-zag profile is observed; (iii) from 80 nm to 200 nm each single indentation can be distinguished.

From Figure 3a–c, it appears that as the step increases, the effect of overlapping between adjacent indentations is diminished more and more, and continuous structures can be fabricated only by taking advantage of the overlapping of neighbour indentations which, in our case, occurs when steps are less than 40 nm.

A quantitative description of the continuity of the grooves can be provided by the mean height of primary profile (P_c_) value, which represents the average value of the height of the curve element along the sampling length [56]. From Figure 3j, where P_c_ values as a function of the step are reported, it can be clearly seen that, with the increase of the step, the P_c_ value increases too. When the P_c_ is comparable with PMMA surface roughness (about 0.5 nm), smooth nanochannels are obtained; while when P_c_ slightly exceeds the R_q_ surface value, zig-zag lines start to appear. This increasing trend is more accentuated for steps greater than 70 nm when the sculpted grooves are a succession of single indentations.

The overlapping of the indentations also affects the depths (d) of the nanogrooves, as shown in Figure 3k. In correspondence to the smallest step (10 nm), the maximum depth of (24.43 ± 0.42) nm is achieved; then, after a rapid decrease up to (14.48 ± 0.48) nm (50 nm step), an almost constant depth value of about 14.5 nm is obtained. Based on the presented results, it can be stated that the maximum step that guarantees a continuous and smooth profile of nanostructures is 40 nm. It is worth noting that this maximum step is about half of the curvature radius of the tip used (for DCP20, the provider reports a curvature radius of 100 nm). The choice of setting a step slightly lower than the half of the tip radius guarantees an effective overlapping of indentations even though the structures are about 40% less deep than the grooves patterned with the shortest step. In any case, another parameter, such as the setpoint, that does not affect the lithography duration, can be opportunely tuned to compensate the observed depth decrease. Additionally, it can be stated that the obtained dotted shape of nanostructures is not necessarily a detrimental aspect of the technique. In fact, this combination of indentation parameters could be exploited for the creation of periodic nanostructures or arrays of holes arranged in an orderly manner on lines, whose distance can be finely adjusted. This could be an alternative way to more conventional nanoindentation methods.

### 3.2. Effect of the Force Applied on the Nanogrooves Shape

The variation in depth, width, and shape of the nanogrooves under different force regimes was then investigated.

The template for this set of experiments was composed of fourteen parallel lines; to each line, an increasing setpoint value from 0.5 nA to 7 nA was assigned. In addition, we chose the minimum step and pulse width provided by our AFM that enable a continuous line profile in the shortest time. Therefore, all the lines were patterned by keeping constant the step and the pulse to 10 nm and 10 ms, respectively. The correctness of nanolithography procedure was monitored using an oscilloscope and the AFM probe deflection signals during the patterning were reported in Appendix A.

The nanolithography results were characterized with AFM with NSG01 tips at high resolution, (Figure 4). As shown in the Figure 4 from a to c, the shape of the nanochannels is highly regular and homogeneous; all the channels are V-shaped (Figure 4d–f) and, at their borders, bulges are asymmetrically accumulated, whose height increases with the increasing of the force applied. We have already observed the accumulation around the structures of the polymer material removed and the asymmetry in its height due to the bending of the cantilever during indentation. We further demonstrated the PMMA bulge removal using both dry [48] and wet [49] etching cleaning processes.

For each nanochannel on PMMA, the depths (d) and widths (w) were evaluated by randomly measuring twenty different cross sections along the patterned grooves on the AFM images. The d and w values were then plotted as a function of setpoint (Figure 4g,h, respectively). As shown in the Figure 4g, the channels’ depth increases as a function of the normal force applied, ranging from (1.17 ± 0.24) nm to (38.68 ± 1.40) nm. The depth increase is not linear in the whole range of the forces applied: with the increase in the setpoint, the tip penetrates more and more into the polymer layer and the contact area between tip and polymer increases too. Consequently, the adhesion and friction forces on the tip become greater, resulting in a reduction of the effective normal force acting on the tip.

As for the depth, with the increase of the force applied, an enhancement of the nanochannels’ width is observed from (52.78 ± 5.11) nm to (165.3 ± 6.67) nm.

The flatness of the nanogrooves was estimated by measuring the 2D R_q_ value in the middle of the nanogrooves. Regardless of the depth of the channels, the roughness values were very low, ranging from (0.3 ± 0.09) nm to (0.99 ± 0.08) nm, comparable to that of pristine PMMA.

### 3.3. Impact of the Pulse Width on the Nanogrooves Shape

The third key nanolithography parameter considered is the pulse width, i.e., how long the piezo scanner takes to extrude in z-direction inducing the tip to penetrate through the PMMA surface for each indentation. As can be easily guessed, this parameter strongly affects the length of nanolithography process in terms of time. Fourteen lines were patterned changing the pulse width from 10 ms to 200 ms, in step of 10 ms up to 100 ms and 25 ms from 100 ms to 200 ms, whereas the force applied was kept at (9.44 ± 0.25) µN (5 nA setpoint), and 10 nm of step, already used.

As previously conducted, the correctness of the P-AFL procedure was monitored by collecting the AFM probe deflection signals during the patterning procedure by means of an oscilloscope; these signals were reported in Appendix A.

The grooves appear well-patterned and surrounded by pileups (Figure 5a–c) as already observed [49]. As performed in the previous sections, the impact of pulse width on the nanolithography results in terms of d, w, and R_q_ was estimated by randomly measuring twenty cross-sections along the nanogrooves’ AFM images; the d, w, and R_q_ mean values were then plotted as a function of pulse width (Figure 5d–f, respectively).

As can be seen in Figure 5d, the d of the nanogrooves rapidly increases from (27.99 ± 0.65) nm to (35.39 ± 1.05) nm for width values of 20 ms and 50 ms, respectively, while no relevant variation in the nanochannels’ depth is observed for pulse widths ranging from 70 ms to 200 ms.

As for the nanochannels’ width, with the increase of the pulses’ width from 10 ms to 60 ms, a relevant increase in nanochannel width, ranging from (135.79 ± 2.31) nm to (150.81 ± 0.66) nm, was observed (Figure 5e); for longer pulses, ranging from 70 ms to 200 ms, a plateau is reached and only a slight increase of line width is recorded: from (151.99 ± 1.35) nm to (160.3 ± 1.51) nm (Figure 5e). The observed trends of both results can be explained by taking into account the elastic recovery of PMMA after nanoindentation. It is useful to remind that the shorter the voltage pulses applied to the AFM piezo scanner, the shorter the duration of the nanoindentation. Then, in correspondence to shorter pulses, the PMMA can better recover, while 100 ms seems like a threshold beyond which no recovery occurs, and a maximum width is reached.

The 2D R_q_ value was estimated in the middle of each nanogrooves and the mean R_q_ value was plotted as a function of pulse width (Figure 5f). As shown in Figure 5f, with the increase pulse width, a slight decrease in R_q_, from (0.63 ± 0.03) nm to (0.33 ± 0.03) nm was observed. The very low R_q_ values, comparable to the roughness of pristine PMMA, clearly indicate that, regardless of the pulse width used, the nanochannels are very smooth.

### 3.4. Tip Wear Test

One of the main advantages of the P-AFL technique is the capability to prevent damage to the AFM tip. The tip’s degradation during the nanolithography process is a critical point for many AFM-based lithography techniques that strongly limits the use and the spread of AFM-based lithographic techniques. For this reason, we observed the DCP20 tip by Scanning Electron Microscope (SEM), prior to (Figure 6a,b) and after the nanolithography testes (Figure 6c,d). The same AFM tip was used to pattern more than one hundred lines, with a total sculpted length of about 1000 micrometres. At first glance, no PMMA debris is attached on the apex of the tip after its usage, and no relevant tip damages are observed. To confirm this observation, ImageJ software was used to measure the curvature tip radius of both new and used DCP20 probes. The radii found were (99.6 ± 3.6) nm and (86.6 ± 8.8) nm, respectively. The slight decrease in the tip radius, together with the observation of no PMMA residues on the tips after numerous lithography tests, allow us to state that the P-AFL prevents tip damage and tip wear.

### 3.5. Fabrication of Complex Unconventional Shape Nanostructures

P-AFL readily allows for the fabrication of arbitrary shapes, sizes, and curvatures of nanostructures with high spatial resolution. In order to demonstrate the capabilities and, thus, the versatility of the method, we patterned a set of unconventionally shaped nanostructures. By exploiting the results of the systematic study on the optimization of the P-AFL parameters (step, pulse width, setpoint), we were able to pattern structures with highly complex shapes and a constant depth profile, such as a polygonal nanolabyrinth (Figure 7a), an array of concentric and circular tranches (Figure 7b), and a serpentine-shape nanofluidic channel (Figure 7c,d). In addition, the nanostructures with a varying depth profiles and complex shapes were patterned:(i)an equilateral triangle with 4 µm side length (Figure 7e);(ii)a circle 4 µm in diameter (Figure 7f);(iii)a serpentine-shape nanochannel, with a ~24.5 µm total length (Figure 7g).

All the nanostructures in Figure 7 were patterned by fixing the step value to 40 nm and the pulse width to 30 ms. For all the nanostructures patterned with CP-AFL, a constant setpoint value, equal to 5 nA, was chosen. The nanostructures with a varying depth profile were patterned with a setpoints linearly increased from 0 to 5 nA.

The examples reported in Figure 7 demonstrate that P-AFL can be used for patterning complex nanostructures, composed by both straight and circular trenches, with high precision and spatial resolution. This is further confirmed by Figure 7d, where a 3D AFM image of the nanofluidic channels is reported. As it can be seen the bend of trench at the vertices of the pattern appears continuous and homogeneous.

## 4. Conclusions

In this work, we presented a careful and accurate investigation of the effect of the main P-AFL parameters on the quality of the nanostructures patterned on a thin PMMA layer. At first, three main nanolithography parameters were selected, i.e., distance between following indentations (step), pulse amplitude, and width. After a systematic modification of the parameters, we measured the variation in the depth, width, and overall shape of the nanochannels. In more detail, when increasing the distance between indentations, the shape of the nanogrooves change drastically, passing from continuous to dotted lines when the step is over 40 nm. Moreover, the increase in the step length affects not only the continuity of the channels but also their depth. This finding is due to the reduction in overlapping between single indentations.

Then, we investigated the variation in the nanogrooves’ morphology under different force regimes. As expected, with the increase in the normal force applied, the depth and the width of the grooves increase too, with an almost linear trend, while the channel roughness values remain almost unchanged.

We further evaluated the impact, on the nanogrooves shape, of the pulse width, i.e., the P-AFL parameter that mainly affects the length of the nanolithography process (in terms of time). We observed that, the increase in pulse width mainly induced enlargement of the nanogrooves while the depth and roughness of the channels remained almost unmodified. In addition, the AFM probes used for the nanopatterning were observed using SEM microscopy to measure the tip degradation degree induced by P-AFL tests. SEM analysis revealed a very slight decrease in the tip curvature radius and no PMMA debris attached to the AFM probes, confirming that P-AFL prevents tip wear.

Finally, we patterned unconventionally shaped nanostructures, such as polygonal nanostructures; a nanolabyrinth; an array of concentric, circular tranches; and a serpentine-shape nanofluidic channel exploiting the potentiality of the technique and the optimization of the P-AFL parameters (step, pulse width, setpoint). This was not a mere skill exercise, but a concrete demonstration of the P-AFL method’s versatility in the patterning of nanostructures with highly complex shapes.

## Figures and Tables

**Figure 1 nanomaterials-12-04421-f001:**
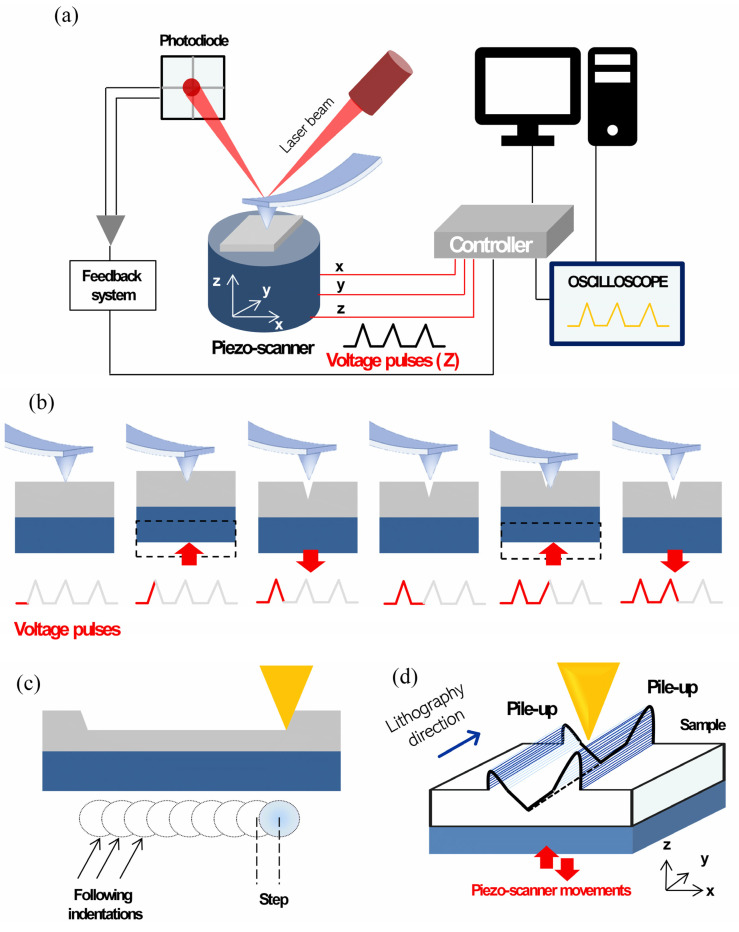
(**a**) sketch representation of AFM NTEGRA instrumentation; (**b**) schematic images, in lateral view, of the P-AFL operation mode: the AFM tip is brought in contact with the sample surface and a train of V-shaped voltage pulses (grey and red V-shaped lines) are applied to the piezoelectric scanner, inducing its movement in z direction. Then, AFM probe indents the polymer surface. After that, a second pulse is applied to the scanner and a second indentation occurs close to the first one; (**c**) lateral view of a nanochannel sculpted by P-AFL and top view of single, overlapped indentations; (**d**) 3D sketch image of the nanolithography process, in which the nanolithography direction is highlighted.

**Figure 2 nanomaterials-12-04421-f002:**
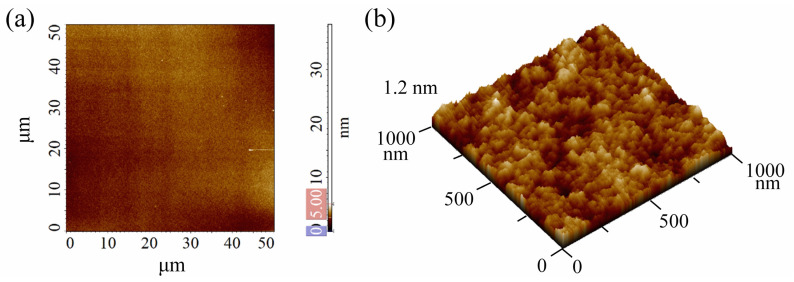
AFM high-resolution images of PMMA surface, acquired on (**a**) (50 × 50) µm and (**b**) (1 × 1) µm areas. The pink and blue boxes on the z-scalebar in (**a**) represent the minimum and maximum adjustment of the colour scale.

**Figure 3 nanomaterials-12-04421-f003:**
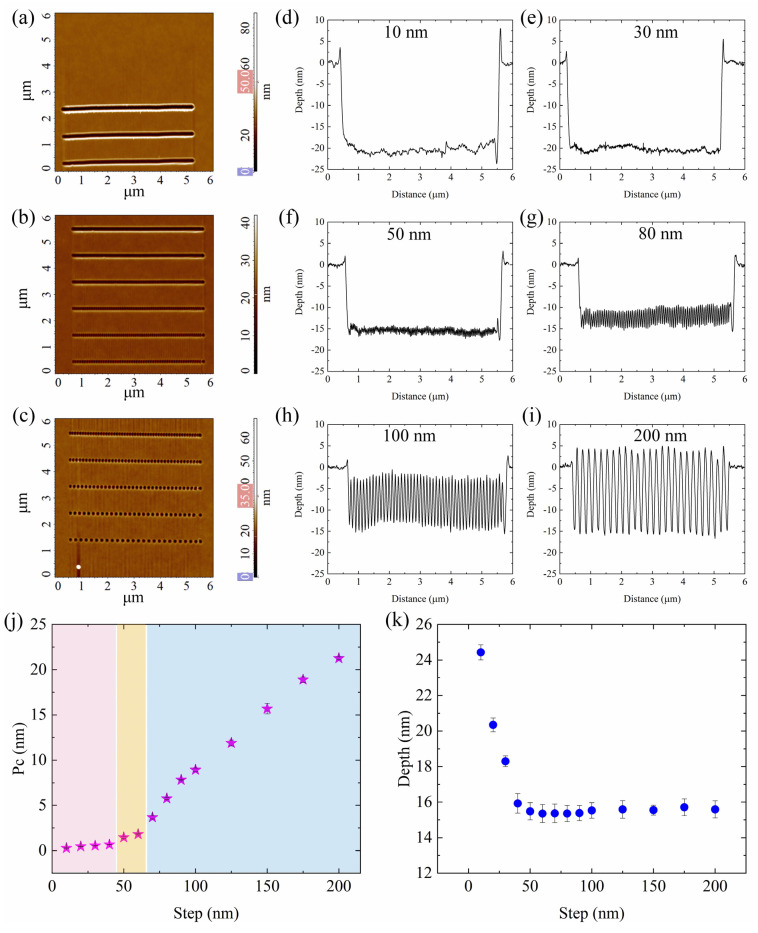
(**a**–**c**) SensHeight AFM images of nanogrooves sculpted with an increasing step values from 10 to 200 nm. The pink and blue boxes on the z-scalebar in (**a**–**c**) represent the minimum and maximum adjustment of the colour scale. (**d**–**i**) cross-sections of nanochannels sculpted with a step of 10, 30, 50, 80, 100, and 200 nm, respectively; (**j**) mean P_c_ and (**k**) depth values, obtained by averaging 3 and 25 measurements, respectively, and plotted as a function of the step. The data reported in (**j**) and (**k**) are statistically significant for *p* < 0.01.

**Figure 4 nanomaterials-12-04421-f004:**
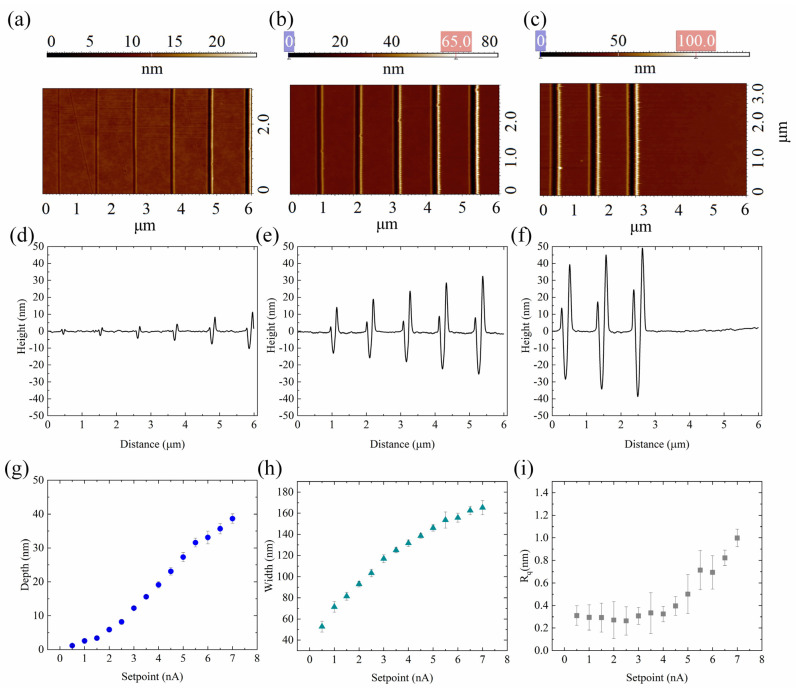
(**a**–**c**) SensHeight AFM images of nanochannels patterned on PMMA with an increasing setpoint value (from 0.5 to 7 nA in step of 0.5 nA) and (**d**–**f**) relative cross section profiles; (**g**–**i**) report the variation of the mean depth, width, and roughness for each fixed setpoint. The nanochannels’ depth is measured from the PMMA surface to the deeper part of the grooves while the width is defined by the full width at the top of the nanochannels. Both d and w were reported as mean values, obtained by averaging their values at twenty-five different locations, randomly chosen from the AFM topographical images. The pink and blue boxes on the z-scalebar in (**b**,**c**) indicate the minimum and maximum adjustment of the colour scale. The data reported in (**g**,**h**) are statistically significant for *p* < 0.001 while the data in (**i**) are statistically significant for *p* < 0.05.

**Figure 5 nanomaterials-12-04421-f005:**
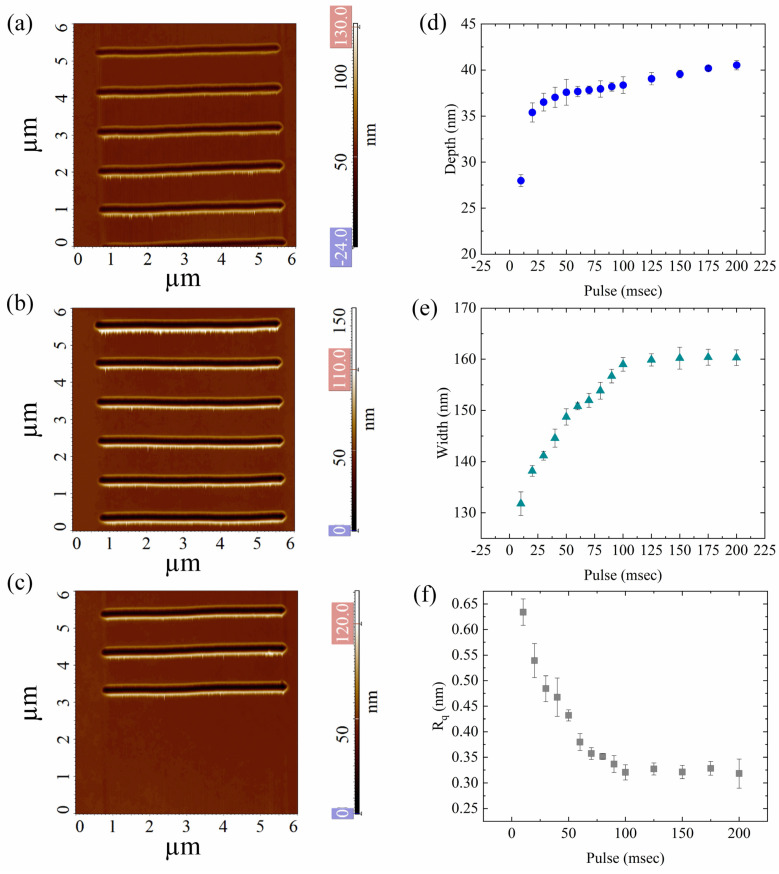
(**a**–**c**) SensHeight AFM topographic images of nanochannels patterned with an increasing pulse width value (from 10 ms to 200 ms); (**d**–**f**) report the variation of the mean depth, width, and roughness of nanochannels patterned with increasing pulse width values. The d, w, and R_q_ values were reported as mean value, obtained by averaging their values at twenty-five different locations, randomly chosen from the AFM acquisitions. The pink and blue boxes in Figure 5a–c represent the minimum and maximum adjustment of the colour scale. The data reported in (**d**,**e**) are statistically significant for *p* < 0.001 while the data in (**f**) are statistically significant for *p* < 0.05.

**Figure 6 nanomaterials-12-04421-f006:**
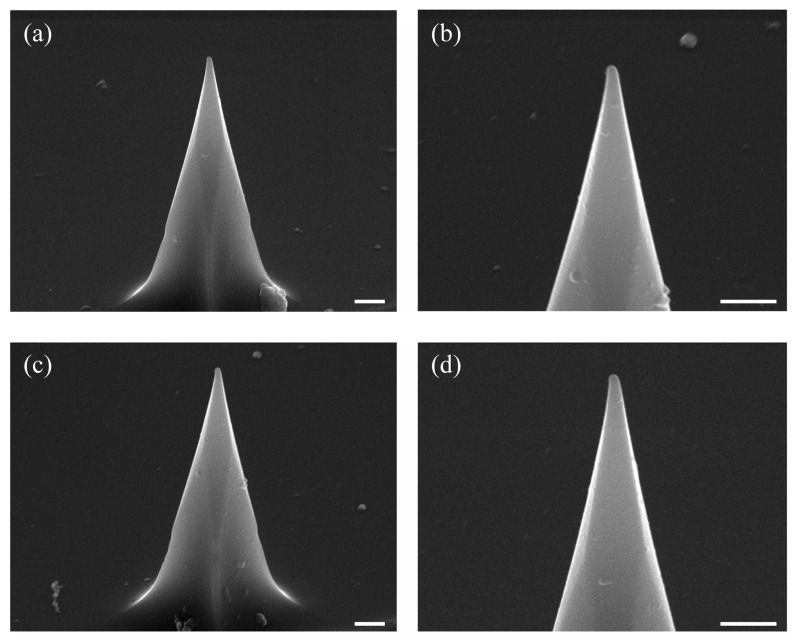
SEM images of (**a**,**b**) a new DCP20 probe and (**c**,**d**) the same probe used for P-AFL tests; the scale bars in the SEM figure correspond to 1 µm.

**Figure 7 nanomaterials-12-04421-f007:**
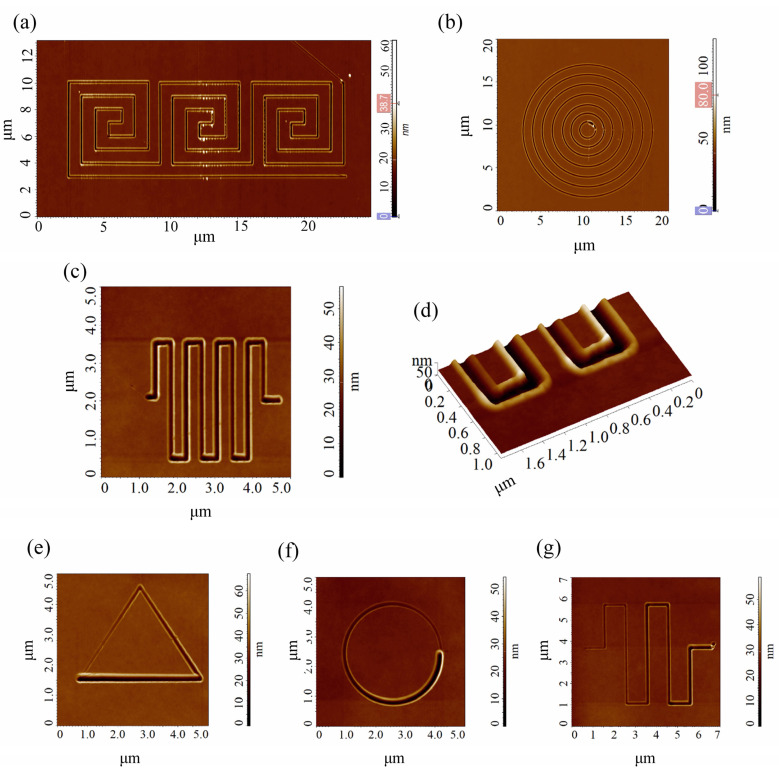
Two-dimensional SensHeigh AFM images of (**a**) nanolabyrinth, (**b**) an array of concentric circular-shape nanostructures, (**c**) a serpentine-like nanofluidic channel and (**d**) 3D SensHeight AFM images of nanofluidic channel loops. All these unconventional-shape nanostructures were fabricated by Constant Pulse-AFL. (**e**–**g**) Two-dimensional AFM images of triangular, circular, and a serpentine-like nanochannels patterned on PMMA by Gradient Pulse-AFL. The pink and blue boxes on the z-scalebar in (**a**,**b**) represent the minimum and maximum adjustment of the colour scale.

**Table 1 nanomaterials-12-04421-t001:** Force values estimation by force spectroscopy.

Setpoint (nA)	0.5	1	1.5	2	2.5	3	3.5	4	4.5	5	5.5	6	6.5	7
**Fz (µN)**	1.04	1.77	2.50	3.36	4.10	4.85	5.88	7.12	8.23	9.44	10.33	10.75	11.67	12.09
**St. Dv. (µN)**	0.06	0.01	0.12	0.12	0.11	0.19	0.60	0.34	0.48	0.25	0.26	0.74	0.38	0.44

## Data Availability

The data supporting this study’s findings are available from the corresponding author upon reasonable request.

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
