# Peer review of "Investigation of the Effects of Pulse-Atomic Force Nanolithography Parameters on 2.5D Nanostructures’ Morphology"

_nanomaterials, 2022, doi:10.3390/nano12244421_

Round 1

Reviewer 1 Report

The manuscript reports some developments in machining SPL to generate nanogrooves on a thin polymer layer (PMMA). The novelty of the approach lies in applying discrete z-displacements (pulses) to generate the nanopatterns. In this way three parameters might be used to control the nanopatterns, the z-displacement and the lateral distance between indentations. Figure 3 illustrates some features of the approach such as the spatial resolution and reproducibility.

This is an interesting contribution that enhances the reproducibility of machining-SPL on PMMA films. It has the quality to be published in Nanomaterials after some revision.

Comments

1 The nanolithography methods based on using an AFM; are called Scanning Probe Lithography (SPL). Please see definitions in R. Garcia et al. Nature nanotechnology 9, 577-587 (2014).

2 I would suggest to modify the title. The term ‘pulse’ is commonly associated with the application of a voltage. In the manuscript, it refers to discrete displacements of the z-piezo of the AFM.

3 Optional suggestion. The authors might consider to cite the review on oxidation SPL by Y. Ryu and R. Garcia, Nanotechnology 28, 142003 (2017).

Author Response

We have reported the answers to the Reviewer's questions in the attached file

Reviewer 2 Report

The authors present comprehensive experimental results. These results are impressive. The manuscript should be accepted after minor revision.

1. The roughness of PMMA should be mentioned.

2. The conclusion part needs to be concise.

Author Response

We have reported the answers to the Reviewer's questions in the attached file.

Reviewer 3 Report

The manuscript "Investigation of the effects of Pulse-Atomic Force Nanolithography parameters on 2.5D nanostructures’ morphology" by Pellegrino et al. describe a more in depth study on their recently published patterning approach (47,48) for optimizing writing parameters to exploit the method for more complex structures.

In general, the manuscript is well written and describes the experiments comprehensively. The experiments are a neat demonstration of the capabilities of their approach for 2.5D patterning of polymer films. In the following a few comments that could be considered for (further) improvement of the manuscript by clarifying a few aspects in more detail:

1. Embedding of the method in the context of various DPL methods:

Generally, I'm a bit puzzled why it is termed a "new" method with new abbreviation and so on, as the overall approach looks still very much lyke dynamic plowing to me, with the main difference, that the dynamics is induced by the piezo stage rather than by the cantilever actuation. Anyway, this is of course a matter of taste. However, given this very close similarities, I would suggest to add a brief discussion (e.g. in section 2.3) that clearly and in one place declares the main differences in the approach. In particular looking e.g. at the recent review by Yan et al. ("Scratch on Polymer Materials Using AFM Tip-Based Approach: A Review", Y. Yan, S. Chang, T. Wang, Y. Geng, Polymers 2019, 11, 1590.), there are very similar approaches also by letting either the stage or the cantilever vibrate. Also (but this would be better done in the conclusions) a brief remark on the key advantages and limitations in comparisson to other DPL approaches should be added.

2. Is there a dependance of outcome of the lithography with writing direction? This is a very common thing in nanoscratching and other tip based methods, and could be discussed e.g. in connection to e.g. Y. He, Y. Yan, Y. Geng, E. Brousseau, Applied Surface Science 2018, 427, 1076. Did the authors observe similar phenomena in their multidirection patterns of Figure 5?

3. Did the authors consider different materials / polymers? E.g. there is pronounced differences in nanoscratching on polymers versus polymer brushes (M. Hirtz, M. K. Brinks, S. Miele, A. Studer, H. Fuchs, L. Chi, Small 2009, 5, 919.) and on different materials (mechanical properties of the polymer). Did the authors also try other polymer configurations or materials as PMMA? Or what would be the prospects for other materials in their method?

4. One of the claims for the advantage of the method is the avoidance of wear to the tip in comparisson to other methods. However, the authors use a diamond coated tip, which is in itself already much more wear resistant. E.g. Hirtz et al. (same ref. as in point 3) inflict even purposful damage to the tip also ending up with a similar size as the diamond tip (100nm stated by the author), to enable stable scratching. Have the authors tried a "regular" AFM tip in their P-AFL approach and what was the performance there. And what would happen with a diamond coated tip in regular DPL? This should be discussed in the section about wear, as to better estimate the (potential) benefits in regard to wear given by the authors approach.

5. What was the writing speed, e.g. what's the total writing time for the different structures in Fig. 7? As speed is another claim for the benefits of the P-AFL approach, this should be discussed somewhere in the manuscript in more detail, as in general I would expect (given the higher masses to be moved of the stage in comparisson to the actuation of the cantilever) that theoretical maximum speed should be possible rather on the side of cantilever actuation. 

6. Minor:

- line 72

"continuously variable depth profile" should maybe better be written as "continuously varying depth profile"? As the authors (in 47) describe grooves with continuoly increasing depth.

- line 463

there is a striked out "on" in the sentences that should be removed.

- In the AFM images of Figs. 2-5 there are colored boxes on the scales. What is their purpose? This should be remarked to in the captions or changed. I gather it is about an adjustment to the min / max for the actual color scale but then better adjust the whole scale.

- the reference (47) gives only authors, title and year, should be "P. Pellegrino, A. P. Bramanti, I. Farella, M. Cascione, V. De Matteis, A. Della Torre, F. Quaranta, R. Rinaldi, Nanomaterials 2022, 12, 991."

Author Response

(The authors gave the same response as above.)
